**Research Article, SciPost Physics**

Dear Editor,

We sincerely thank you for considering our manuscript entitled *"Spin-Orbit Photonics in a Fixed Cavity: Harnessing Bogoliubov Modes of a Bose–Einstein Condensate"* for publication in *SciPost Physics*, and for forwarding the referee reports to us. We are also grateful to the referees for their thoughtful evaluations, insightful suggestions, and constructive feedback.

Their comments have significantly contributed to improving the quality and clarity of our manuscript. We deeply appreciate their efforts in helping us strengthen our work.

In the revised version of the manuscript, we have addressed all of their concerns comprehensively. We believe that the updated manuscript now meets the standards for publication in *SciPost Physics*, and we respectfully submit it for your further consideration.

Below we provide detailed, point-by-point responses to all referees' comments. We also wish to inform you that, in the revised manuscript, **Ghaisud Din** is listed as the first author, with the consent of all co-authors, in recognition of his substantial contribution to the revision. Referees' comments are shown in indented blue font, and our responses are provided in non-indented black font. All major changes in the revised manuscript are marked in bold for clarity.

We also appreciate your efforts in handling our manuscript, and we look forward to receiving your final decision.

Thank you.

On behalf of all authors.

<b>Response to Referee 1</b>

> The authors Muqaddar Abbas et al. study in the theoretical work with the title "Spin-Orbit Photonics in a Fixed Cavity: Harnessing Bogoliubov Modes of a Bose–Einstein Condensate" the photonic spin-Hall effect modified by an optomechanical system. In particular, they investigate the linear response theory of a driven cavity that couples to a mechanical mode of a Bose-Einstein condenstate and use the result for the susceptibility to obtain reflection coefficients and spin-dependent displacements of incident electric field with different polarizations. They visualize their results for various experimentally realistic parameters and

> highlight how the spin-selective scattering can be modified by changing model parameters.

We thank the reviewer for the careful and concise summary of our work. We fully agree with this description and appreciate the clear understanding of the main contributions of our study.

> In my opinion this is an interesting and timely work and the combination of the photonic spin hall effect with optomechanical cavities in this specific situation is also novel. That said I believe that the needed acceptance criteria for SciPost Physics Core are potentially fulfilled. The main weakness, in my opinion, are several short comings in explaining the derivations and the results. This is why I can only fully recommend this draft for publication after the points below have been clarified.

We thank the reviewer for the positive and encouraging comments on our work. We appreciate the constructive feedback regarding the clarity of derivations and results, and we have addressed all the points raised to improve the presentation and explanations in the revised manuscript.

> (1) When the authors introduce the pump they do not specify the polarization nor the angle. Regarding Fig. 1, I believe the angle is 0. Could the authors add this information?

We thank the reviewer for pointing this out. The pump polarization and angle are indeed zero in our calculations.

We have now added this clarification in the revised manuscript text (paragraph introducing the pump), Section II paragraph 1 last three lines on page 2, and updated the caption of Fig. 1.

> (2) In Sec. II there are several quantities undefined or defined much later. In general I would recommend that the authors should massively improve Sec. II because at the moment it is hard to follow and almost impossible to reproduce. It would help if the authors spend more time in explaining the equations. For instance: (a) it is unclear where the transfer matrix formalism is used. (b) the permitivity is only defined later in the section (c) all $y_{mn}$ and $p_{mn}$ are not defined as far as I can see. (d) Can the authors give a formula for the total field outside of the cavity incoming and reflected. I think that would make it clearer what $R_s$ and $R_p$ are. (e) It is also not really clear for me that once I have $\chi$, how would I calculate $R_s$ and $R_p$? With the transfer matrix or the formulas (3) and (4)?

⁶²     We thank the reviewer for this constructive and detailed comment. In the revised
⁶³ manuscript, we have substantially revised Sec. II—expanding Sec. II.A and Sec. II.B—on
⁶⁴ pp. 2–3 to enhance clarity and reproducibility. Our point-by-point response is provided
⁶⁵ below also

⁶⁶     (a) Use of the transfer–matrix formalism. We now state explicitly where it is used: Sec. II
⁶⁷ introduces the layer transfer matrices in Eq. (1) and forms the total matrix in Eq. (2) by
⁶⁸ multiplying the individual layer matrices. The reflection amplitudes are then obtained from
⁶⁹ the elements of the total matrix via Eqs. (3)–(4), which is where the formalism directly
⁷⁰ yields $R_s$ and $R_p$.

⁷¹     (b)Definition of the permittivity. The permittivity of the BEC layer is now defined at the
⁷² start of Sec. II, immediately after introducing the transfer matrix in Eq. (1). In particular,
⁷³ once the susceptibility $\chi$ is computed, the effective permittivity used in the transfer matrices
⁷⁴ is

$$\varepsilon_2 = 1 + \chi.$$

⁷⁵     (c)Definitions of $y_{mn}$ and $p_{mn}$. These quantities are now defined at first use. Specifically,
⁷⁶ if the total transfer matrix for $s$ (TE) polarization is

$$y \equiv \begin{pmatrix} y_{11} & y_{12} \\ y_{21} & y_{22} \end{pmatrix},$$

⁷⁷ then $y_{mn}$ denotes its $(m,n)$ element. Analogously, for $p$ (TM) polarization we write

$$p \equiv \begin{pmatrix} p_{11} & p_{12} \\ p_{21} & p_{22} \end{pmatrix},$$

⁷⁸ with $p_{mn}$ the corresponding elements. The text now states these definitions explicitly.

⁷⁹     (d)Total field outside the cavity and meaning of $R_s$, $R_p$.

⁸⁰     We place the first mirror $M_1$ at $z = 0$ and work in the half–space $z < 0$ (outside the
⁸¹ cavity). Decompose the incident plane wave at angle $\theta_i$ into TE/TM (s/p) components,
⁸² with unit polarization vectors $\hat{\mathbf{e}}_s$ and $\hat{\mathbf{e}}_p$. Let $\mathbf{k}_\parallel = (k_x, k_y, 0)$ be the in–plane wave vector
⁸³ and $k_{z0} > 0$ the longitudinal component in the external medium ($\varepsilon_0$).

$$\mathbf{E}_{\mathrm{inc}}(\mathbf{r}) = E_0(E_s\,\hat{\mathbf{e}}_s + E_p\,\hat{\mathbf{e}}_p)\,e^{i(\mathbf{k}_\parallel \mathbf{r}_\parallel + k_{z0}z)}, \qquad \mathbf{r}_\parallel = (x, y, 0). \tag{1}$$

⁸⁴     The reflected field in the same half–space is

$$\mathbf{E}_{\mathrm{ref}}(\mathbf{r}) = E_0(R_s\,E_s\,\hat{\mathbf{e}}_s + R_p\,E_p\,\hat{\mathbf{e}}_p)\,e^{i(\mathbf{k}_\parallel \mathbf{r}_\parallel - k_{z0}z)}. \tag{2}$$

Hence, the *total external field just outside the cavity* is

$$\mathbf{E}_{\text{tot,out}}(\mathbf{r}) = \mathbf{E}_{\text{inc}}(\mathbf{r}) + \mathbf{E}_{\text{ref}}(\mathbf{r}) \tag{3}$$

which makes explicit that $R_s$ (TE) multiplies the s–component and $R_p$ (TM) multiplies the p–component of the reflected wave.

In our transfer–matrix model, $R_s$ and $R_p$ are obtained from the total matrix $y = M_1 M_2 M_3$ (elements $y_{mn}$), yielding the TE/TM Fresnel coefficients

$$R_s = \frac{q_{1s}(y_{11} + y_{12}q_{3s}) - (q_{3s}y_{22} + y_{21})}{q_{1s}(y_{11} + y_{12}q_{3s}) + (q_{3s}y_{22} + y_{21})}, \tag{4}$$

$$R_p = \frac{p_{1m}(y_{11} + y_{12}p_{3m}) - (p_{3m}y_{22} + y_{21})}{p_{1m}(y_{11} + y_{12}p_{3m}) + (p_{3m}y_{22} + y_{21})}, \tag{5}$$

with $q_{is} = \sqrt{\varepsilon_i k_0^2 - k_\parallel^2}$ and $p_{im} = q_{is}/\varepsilon_i$ (the TM counterparts). The intracavity BEC enters through $\varepsilon_2(\omega) = 1 + \chi(\omega)$, so that $R_s$ and $R_p$ inherit their frequency and angle dependence from $\chi$.

For completeness, in the circular basis $\hat{\mathbf{e}}_\pm = (\hat{\mathbf{e}}_s \pm i\hat{\mathbf{e}}_p)/\sqrt{2}$, the reflected $\sigma^\pm$ fields are simply the projection of $\mathbf{E}_{\text{ref}}$ onto $\hat{\mathbf{e}}_\pm$; the Gaussian–beam expression used to analyze the PSHE (our Eq. (1)) follows directly from this decomposition.

(e)How to compute $R_s$ and $R_p$ once $\chi$ is known. First update the BEC-layer permittivity via $\varepsilon_2 = 1 + \chi$. Then build the layer matrices $M_i$ [Eq. (1)] with this $\varepsilon_2$, form the total matrix $y = M_1 M_2 M_3$ [Eq. (2)], and finally evaluate the reflection amplitudes directly from the elements of $y$ using Eqs. (3)–(4). The revised text now includes a short, step-by-step description of this workflow.

We believe these changes greatly improve the readability of the section and thank the reviewer for these useful suggestions.

(3) In Sec. III when the authors introduce the optomechanical system it would be nice to include a few more details. For instance: (a) spontaneous emission is neglected because of far detuning? (b) What about the polarization of the light in the cavity? Where is the magnetic field? Why is it okay to only consider one cavity mode? Why only one internal state of the BEC? (c) There is a damping rate for the mechanical mode introduced? Where does that come from? (d) What about ultra-cold collisions? (e) There is a detuning $\Delta_p$ introduced and I do not understand why? Is it related to $\Delta_p$?

We thank the reviewer for the suggestions to clarify the optomechanical model in Sec. III. We address each sub-point below and indicate explicitly how we have revised the manuscript.

(a) Neglecting spontaneous emission: The probe and pump lasers are far-detuned from the atomic resonance, which ensures that the population of the excited states remains negligible. As a result, spontaneous emission can be safely neglected, and the excited electronic states are adiabatically eliminated. Manuscript update: Added in Sec. III:

"The lasers are far-detuned from the atomic resonance, ensuring that spontaneous emission is negligible and the excited electronic states can be adiabatically eliminated."

(b) Polarization, magnetic field, and cavity mode: We consider a single linearly polarized cavity mode along the $x$-axis. The magnetic field component is included implicitly but does not influence the one-dimensional optomechanical dynamics. Only one internal state of the BEC is considered because other hyperfine states are far-detuned and thus remain negligibly populated. The single-mode approximation is justified as the cavity is operated in a regime where the mode spacing exceeds both the cavity linewidth and the mechanical frequency. **Manuscript update:** Added in Sec. III:

"We consider a single linearly polarized cavity mode along the $x$-axis. The magnetic field component is implicitly included but does not affect the one-dimensional optomechanical dynamics. Only one internal state of the BEC is considered, as other hyperfine states are far-detuned and thus negligibly populated."

(c) Mechanical damping rate: The mechanical mode experiences damping at a rate $\gamma_m$, which accounts for decoherence due to atomic collisions and residual coupling to the thermal environment. **Manuscript update:** Added in Sec. III:

"The mechanical mode is assigned a damping rate $\gamma_m$, which accounts for decoherence due to atomic collisions and residual coupling to the thermal environment."

(d)Ultra-cold collisions: Collisional interactions among the ultra-cold atoms modify the effective mechanical frequency via mean-field contributions. These interactions are incorporated into the system parameters used in our simulations. **Manuscript update:** Added in Sec. III:

"Ultra-cold atomic collisions are incorporated in the effective mechanical fre-
quency via mean-field interactions, which are captured in the system parameters
used in our simulations."

(e)Detuning $\Delta_p$: The detuning $\Delta_p$ corresponds to the effective probe detuning relative to the mechanical Bogoliubov mode frequency $\omega_m$, defined as $\Delta_p = \delta - \omega_m$. This arises naturally in the resolved-sideband regime. **Manuscript update:** Added in Sec. II after Eq. (25) on page 4:

"In the resolved sideband regime, where $\omega_m \gg \kappa$, the effective probe detuning is $\Delta_p = \delta - \omega_m$, which corresponds to the detuning reduced by the mechanical Bogoliubov mode frequency."

We have added clarifications regarding detuning and spontaneous emission, cavity polarization, single-mode approximation, mechanical damping, ultra-cold collisions, and the definition of $\Delta_p$. All additions are placed in Sec. III immediately after introducing the relevant components of the system on page 4 second and third paragraph, improving both clarity and completeness.

(4) I found some small things in Sec. IV that are unclear (maybe there are more) (a) the authors write photonic PSHE but so far they used photonic SHE. Maybe use PSHE everywhere? (b) There is a Figure ?? ". I believe it should be 3? (c) I did not find the definition of $\Delta_p$?

We thank the reviewer for pointing out these unclear points in Sec. IV.

(a) the authors write photonic PSHE but so far they used photonic SHE. Maybe use PSHE everywhere? We agree with the reviewer and have now used the notation "PSHE" (photonic spin Hall effect) consistently throughout the manuscript.

(b)There is a Figure ?? ". I believe it should be 3? The reference "Figure ??" was indeed a typographical error and has been corrected to "Figure 3" in the revised version.

(c)I did not find the definition of $\Delta_p$? The definition of the detuning $\Delta_p$ has now been added in Sec. II when it first appears after equation (25) and is referenced in Sec. IV for clarity.

We thank the reviewer again for these helpful suggestions.

(5) Regarding the description of Figure 3 and 4. There are a few things that are unclear to me. (a) I do not understand the sentence [...] indicating a baseline scenario where spin-

We thank the reviewer for these thoughtful questions regarding the description of Figs. 3 and 4. We have revised the corresponding text to clarify the physical meaning and to remove any ambiguity. In particular,

(a)I do not understand the sentence [...] indicating a baseline scenario where spin-dependent reflection is balanced. Why is it balanced? Also do the authors always choose $\Delta = \omega_m$ here? Is that needed for the symmetry in $\Delta_p$?

We have removed the phrase "indicating a baseline scenario where spin-dependent reflection is balanced" and replaced it with a clearer description of the initial reference point. We now explicitly state that the case $\Delta = \omega_m$ is used throughout Fig. 3 in order to provide a symmetric reference point in the probe detuning, corresponding to the resolved–sideband condition in which the probe is tuned to the mechanical resonance.

**Response to points (b)–(d):**

We thank the referee for these helpful questions. Our original wording was indeed ambiguous and we have revised the text to improve clarity. Below we clarify the intended meaning and indicate the specific changes made in the manuscript.

(b)I also did not understand[...] becomes increasingly negative, a pronounced asymmetry develops in the angular profile of the ration. [...] sharply increases near resonance , showing enhanced [...]. What is meant by asymmetry? Just that there is a peak in $\theta$? What means near resonance? I was thinking about a frequency but maybe the authors are refering to $\theta$?

Meaning of "asymmetry" and "near resonance": By "asymmetry" we do *not* refer to the mere presence of a peak in the $\theta_i$–dependence, but to the fact that, as the probe detuning becomes more negative, the shape of the peak is no longer symmetric with respect to its maximum. Specifically, the slope for $\theta_i < 60.2°$ becomes steeper than that for $\theta_i > 60.2°$. The term "near resonance" was intended to indicate that the pronounced change in the SHE shift occurs close to the critical incident angle $\theta_i \approx 60.2°$, where the cavity–BEC hybrid mode is resonant with the probe. To avoid confusion with frequency resonance, we now write "close to the critical incident angle $\theta_i \approx 60.2°$" throughout the manuscript.

We have updated the manuscript for $\Delta_p = 0$, the angular dependence of the SHE shift displays a symmetric peak centered around the critical incident angle $\theta_i \approx 60.2°$. As the probe detuning becomes more negative, this peak becomes increasingly asymmetric, with a steeper rise for $\theta_i < 60.2°$ and a slower decay for $\theta_i > 60.2°$.'

(c)In the description of Fig.4 the authors write [...] a symmetric Lorentzian-like peak is observed [...]. I do not see any Lorentzian-like peak in Fig. 4. What are the authors refering to? Meaning of "Lorentzian–like peak" in Fig. 4: We agree with the referee that the term "Lorentzian–like" may be misleading. What we intended to describe is that the peak is approximately symmetric for $\Delta_p = 0$. To avoid ambiguity, we have removed "Lorentzian–like" and replaced it with "symmetric peak".

We have updated the manuscript *"When $\Delta_p = 0$ (red dashed line), a symmetric peak is observed, which reflects strong spin–dependent angular deflection due to enhanced light–matter interaction at zero detuning."*

(d)The authors later write [...]the probe detuning becomes increasingly negative [...] the magnitude of the SHE shift increases, but the peak structure becomes increasingly asymmetric and broadened.[...]" I am confused by this statement. What peaks should I be looking at? The peaks in $\theta$ become sharper for more negative $\Delta_p$, correct? Also the peaks are very asymmetric from the beginning for all $\Delta_p$. Sorry for my confusion, probably this is just a misunderstanding.

The "peak structure" refers to the maximum in the SHE shift as a function of the incident angle. While it is correct that this peak becomes sharper as $\Delta_p$ becomes more negative, its *shape* also becomes increasingly asymmetric, as described in (b). We have therefore rephrased the text to avoid suggesting that the peak is broadened.

We have updated the manuscript *"As the probe detuning becomes more negative ($\Delta_p < 0$),*

*the magnitude of the SHE shift increases and the angular peak becomes steeper, while its*
*profile gradually develops an asymmetry with respect to the critical incident angle."*

We hope that these clarifications resolve the referee's concerns. The revised passage in the manuscript now reads: "In Figure 4(a), the normalized PSHE shift exhibits a clear peak at the critical incident angle $\theta_i \approx 60.2°$. For $\Delta_p = 0$, this peak is symmetric. As the probe detuning becomes increasingly negative, its magnitude grows and the peak becomes progressively asymmetric, with a steeper rise for $\theta_i < 60.2°$ than for $\theta_i > 60.2°$. This behaviour reflects the detuning-induced modification of the optomechanical dispersion and the corresponding variation of the spin–orbit coupling strength."

We believe these revisions improve the clarity of the description of Figs. 3 and 4 and we thank the reviewer for pointing out these ambiguities.

(6) Regarding Fig. 5 I have a very minor question. I was wondering why the authors choose $\Delta_p = -0.12\omega_m$? In the caption I read that this corresponds to the maximum photonic SHE. Is that obvious and parameter independent? I mean does this value not depend on $G_{BC}$?

We thank the reviewer for this insightful question. In Fig. 5 we choose the value $\Delta_p = -0.12\,\omega_m$ because, for the parameter set used in this figure, this value corresponds approximately to the maximum of the photonic SHE shift. We agree that this is not parameter independent — in particular, the position of the maximum also depends on the coupling strength $G_{BC}$. We have added a sentence in the revised caption to clarify that the choice $\Delta_p = -0.12\,\omega_m$ is made for the specific parameter set of Fig. 5 and that the exact value of the maximal PSHE shift may shift as $G_{BC}$ is varied.

We sincerely thank the reviewer for their positive assessment of our manuscript. We have carefully revised the manuscript in accordance with the suggested modifications and performed a thorough proofreading to improve clarity, consistency, and presentation. We believe the revised version meets the publication standards and respectfully submit it for final consideration.

## Response to Referee 2

The manuscript discusses the photonic spin Hall effect in a BEC confined within a cavity. As theoretically studied by Risch et al and experimentally shown by Esslinger et al. a the coupling of the cavity field with the BEC Bogoliubov modes mimics an optomechanical

coupling. Such dynamics modulates the photonic spin Hall effect and can be detected by mesuring the field susceptibility. The problem is of timely interest and the result obtained are as far as I can judge correct and new.

We thank the reviewer for the positive assessment and for highlighting the optomechanical analogy in cavity–BEC systems. In the revision, we have added to the fifth paragraph of the *Introduction page 1* (final lines) the sentence: "In cavity–BEC platforms, the coupling between the quantized cavity field and Bogoliubov excitations maps onto an effective optomechanical interaction, as established in theory and reviewed in Ref. [1]. Seminal experiments by the Esslinger group confirmed this paradigm, revealing cavity-enhanced backaction and many-body dynamics [2, 3]." We further clarify how the Bogoliubov-mode–induced susceptibility $\chi$ modifies $\epsilon_2 = 1 + \chi$, which feeds into the transfer–matrix calculation of $R_{s,p}$ and thus the PSHE displacement $\delta_{p\pm}$ (Secs. II.A–II.C). We also note that $\chi$ is experimentally accessible in our readout ($\chi = E_T$, Sec. III). Finally, Sec. II.D articulates the novelty: unlike passive platforms, our cavity–BEC architecture provides *coherently tunable* PSHE via Bogoliubov-mode control.

The manuscript could be published once the authors have expanded section two. The physics of the photonic spin Hall effect is too sketchy and this makes the manuscript difficult to follow. The desction of the effective Bogoliubov modes is somehow clearer – at least to me as I am familiar with the BEC in cavity dynamics. In general the physicalò setup is rather poorly described. They should furthermore put in a clearer perspective the novelty of their research and of their approach.

We thank the reviewer for the constructive feedback. We have revised the manuscript accordingly, with a particular focus on expanding Section II to clarify the physics of the photonic spin Hall effect (PSHE) and the physical configuration. Concretely, we:

(1) Expanded Section II into four parts: II.A Photonic spin Hall effect: physical picture and working formulae, II.B Physical configuration (geometry and conventions), II.C From susceptibility to Fresnel coefficients and PSHE, and II.D Perspective and novelty. (2) Kept the original bold statements in Section II exactly as in the previous version, as requested by the authors' internal constraints. (3) Upgraded the Fig. 1 caption to label beams, angles, and polarization bases; and added Table 1 listing symbols/parameters (values/units match those used in simulations). (4) Added explicit links from the BEC susceptibility $\chi$ to $\epsilon_2 = 1 + \chi$, the transfer–matrix calculation of $R_s$ and $R_p$, and the PSHE displacement $\delta_{p\pm}$. (5) Polished

wording, defined symbols at first use, and corrected minor typographical errors.

1. **"Expand Section two; the physics of the photonic spin Hall effect is too sketchy."**

**Response.** We substantially expanded Section II to make the PSHE physics self-contained before the transfer–matrix results. We now provide an intuitive explanation of how the Fresnel coefficients $R_s$ and $R_p$ lead to spin-dependent centroid shifts, and state the assumptions (paraxial, small-shift limit) explicitly.

**Changes in the manuscript.**

- Added Subsection II.A explaining PSHE and presenting the standard working expressions used in our study and summarized them in Table 1.

- Stated the small-displacement and paraxial assumptions adjacent to the equations.

2. **"The description of the effective Bogoliubov modes is clearer, but the physical setup is poorly described."**

**Response.** We clarified the optical geometry, polarization conventions, and how the pump/probe fields are used in the model. We also made the figure caption more informative.

**Changes in the manuscript.**

- Added Subsection II.B detailing the *Physical configuration (geometry and conventions)*.

- Upgraded Fig. 1 caption to label $\theta_{\text{inc}}$, TE/TM axes, circular bases $\sigma^{\pm}$, and pump/probe paths.

- Added Table 1 listing symbols, definitions, and units/nominal values (matching the simulations).

**3.** **"Put in a clearer perspective the novelty of their research and of their approach."**

**Response.** We now explicitly position our contribution relative to prior PSHE work. We stress that, unlike passive interfaces or metasurfaces, our cavity–BEC platform provides a *coherently tunable* route to control PSHE through the Bogoliubov-mode–mediated susceptibility $\chi$.

**Changes in the manuscript.**

- Added Subsection II.D (Perspective and novelty) highlighting how tuning $\chi$ modifies $\epsilon_2 = 1 + \chi$, thereby reshaping $R_{s,p}$ and the angular dispersion that set $\delta_{p\pm}$.

**4.** **"Link between the BEC susceptibility and the optical response."**

**Response.** We made the modeling chain explicit: $\chi \rightarrow \epsilon_2 = 1 + \chi \rightarrow$ layer matrix $\mathcal{M}_2$ $\rightarrow$ total matrix $y \rightarrow R_s, R_p \rightarrow \delta_{p\pm}$.

**Changes in the manuscript.**

(1)Added Subsection II.C *From susceptibility to Fresnel coefficients and PSHE* to bridge Section II with the susceptibility derived in the following section.

(2) Section II restructured into II.A–II.D with added context and explicit symbol definitions. (3) Fig. 1 caption expanded; Table 1 added. (4) Preserved the authors' original bold text in Section II verbatim.

**We hope the revisions address the concerns and thank you for the positive assessment regarding publication upon revision.**

[1] H. Ritsch, P. Domokos, F. Brennecke, and T. Esslinger, Rev. Mod. Phys. **85**, 553 (2013).

[2] F. Brennecke, S. Ritter, T. Donner, and T. Esslinger, Science **322**, 235 (2008).

[3] K. Baumann, C. Guerlin, F. Brennecke, and T. Esslinger, nature **464**, 1301 (2010).

# Spin-Orbit Photonics in a Fixed Cavity: Harnessing Bogoliubov Modes of a Bose–Einstein Condensate

Ghaisud Din,[1] Muqaddar Abbas,[1, *] and Pei Zhang[1, †]

[1]*Ministry of Education Key Laboratory for Nonequilibrium Synthesis and Modulation of Condensed Matter,*
*Shaanxi Province Key Laboratory of Quantum Information and Quantum Optoelectronic Devices,*
*School of Physics, Xi'an Jiaotong University, Xi'an 710049, China*

We present a theoretical investigation of spin-orbit photonics within a fixed mirror cavity system containing a Bose-Einstein condensate (BEC), in which the Bogoliubov excitation modes of the condensate are treated as effective mechanical oscillators. By embedding the condensate in a single-mode optical cavity, we explore the emergence and modulation of the photonic spin Hall effect (PSHE) through the spin-dependent transverse shifts of a weak probe field. The optical response—encoded in the real and imaginary components of the output field susceptibility—is systematically analyzed as a function of the condensate–cavity coupling strength, revealing a controllable enhancement or suppression of the spin-orbit interaction. Our model captures how the interplay between collective BEC excitations and cavity photon dynamics induces nontrivial modifications in spin-dependent light propagation. Notably, we uncover that the Bogoliubov mode coupling acts as a tunable channel for mediating spin angular momentum transfer within the cavity, offering a novel route for engineering compact, quantum-coherent spin-orbit photonic devices.

## I. INTRODUCTION

The photonic spin Hall effect (PSHE) is a prominent manifestation of spin–orbit coupling (SOC) in optical systems, giving rise to a polarization-dependent lateral displacement of light at material interfaces [1]. This spatial splitting occurs due to the intrinsic interaction between the spin angular momentum of photons (associated with polarization) and their orbital trajectory, paralleling the spin Hall effect in electronic systems [2]. In such a photonic analogue, variations in the refractive index emulate the role of an electric field, producing a transverse shift that depends on the circular polarization components of the incident beam [1, 3, 4].

The theoretical basis for the PSHE was first established by Onoda *et al.* [1], with subsequent advancements contributed by Bliokh and co-authors, who offered a deeper understanding through conservation laws and geometric phase considerations [3]. Experimental verification was achieved in 2008 by Hosten and Kwiat via weak measurement protocols, enabling direct observation of the tiny polarization-induced beam shifts [5].

At its core, the PSHE arises from the conservation of total angular momentum in light–matter interactions, serving as a vital probe of spin–orbit phenomena in structured and inhomogeneous media [2, 6]. Techniques such as weak value amplification have proven highly effective in detecting the subtle spin-dependent displacements, significantly boosting the sensitivity of optical measurements [7, 8]. The broad utility of this effect has made it a cornerstone for numerous applications, ranging from precision metrology and quantum photonic devices to high-resolution optical imaging and sensor development [9–11].

Recent advancements in light–matter interaction have laid the foundation for the development of highly controllable quantum optical platforms, enabling precise manipulation of photonic and atomic degrees of freedom [12–15]. Within this context, Bose-Einstein condensate (BECs) have emerged as exemplary quantum many-body systems, exhibiting macroscopic coherence and distinct superfluid properties that set them apart from conventional phases of matter [16]. The experimental realization of BECs in ultracold dilute gases has catalyzed major progress in quantum optics and quantum simulation [17, 18], offering deep insights into quantum phase transitions, long-range coherence, and collective excitations [19–21].

The integration of BECs with high-finesse optical cavities has opened new frontiers for investigating hybrid quantum systems, where the collective motion of ultracold atoms strongly couples to quantized cavity modes [22, 23]. These platforms allow the exploration of self-organized phases [24], cavity-mediated superradiance [25], and nontrivial optical nonlinearities in the few-photon regime [26]. Notably, the condensate's low-energy excitation spectrum—particularly the Bogoliubov modes—exhibits behavior analogous to mechanical oscillators [27–29]. These quasi-particle excitations, being sensitive to cavity field fluctuations, introduce a new optomechanical degree of freedom that is both tunable and coherent. **In cavity–BEC platforms, the coupling between the quantized cavity field and Bogoliubov excitations maps onto an effective optomechanical interaction, as established in theory and reviewed in Ref. [22]. Seminal experiments by the Esslinger group confirmed this paradigm, revealing cavity-enhanced backaction and many-body dynamics [16, 24].**

Consequently, the BEC can be viewed not only as a quantum fluid but also as a dynamic mechanical element in cavity optomechanics. This dual nature en-

---

* muqaddarabbas@xjtu.edu.cn
† zhangpei@mail.ustc.edu.cn

ables light-induced manipulation of condensate excitations and reciprocal modulation of the intracavity field. The bidirectional coupling between the BEC and the photonic field has been corroborated by theoretical and experimental investigations, revealing significant modifications to the system's optical susceptibility and quantum noise characteristics [16, 30]. In this work, we harness these properties to study spin-orbit photonic phenomena—specifically, the PSHE—within a static cavity-BEC system, wherein Bogoliubov modes serve as mechanically active channels for mediating spin-dependent light propagation.

In this work, we theoretically investigate a hybrid cavity quantum electrodynamics system in which a BEC, confined within a fixed-mirror optical cavity, serves as an effective mechanical oscillator via its Bogoliubov excitation modes. Our focus lies in exploring how the spin–orbit interaction of light, manifested through the PSHE, is influenced by the dynamical coupling between the quantized intracavity field and the collective modes of the condensate. By analyzing the real and imaginary parts of the output probe field susceptibility, we characterize the spin-dependent light shifts induced by this coupling and elucidate the conditions under which spin-orbit photonic effects can be enhanced or suppressed. The role of the effective optomechanical interaction strength is examined in detail, revealing its impact on the PSHE signature. This study provides new insight into the coherent control of spin–orbit photonic phenomena using quantum fluids of light and matter, offering potential avenues for the realization of tunable, compact spin-sensitive photonic devices based on atomic condensates.

## II. Theoretical Model and Physical Configuration

We consider a hybrid optical system consisting of a BEC of $N$ ultracold $^{87}$Rb atoms confined inside an optical resonator, as shown in Fig. 1. A weak probe laser field, incident at an angle $\theta_{\text{inc}}$ on the partially reflective mirror $M_1$, interrogates the system. The probe beam has both transverse electric (TE) and transverse magnetic (TM) polarization components, with frequency $\omega_{\text{P}}$ and power $P_{\text{P}}$. The electric field amplitude is given by $|E_{\text{P}}| = \sqrt{\frac{2\kappa P_{\text{P}}}{\hbar\omega_{\text{P}}}}$, where $\kappa$ denotes the cavity decay rate. Additionally, the cavity is coherently driven by a pump laser of frequency $\omega_{\text{L}}$ and power $P_{\text{L}}$, with amplitude $|E_{\text{L}}| = \sqrt{\frac{2\kappa P_{\text{L}}}{\hbar\omega_{\text{L}}}}$. **In all simulations, we consider a normally incident pump with zero-degree polarization angle.**

Upon reflection, the orthogonal circular polarization components of the probe field undergo a spatial separation transverse to the plane of incidence, indicative of the PSHE, as illustrated in Fig. 1. To make the manuscript self-contained, we first summarize the physical picture of the photonic spin Hall effect and how it connects to our transfer–matrix model and to the BEC-induced susceptibility.

### II.A Photonic spin Hall effect: physical picture and working formulae

The PSHE in reflection originates from a spin–orbit interaction of light: the complex Fresnel coefficients $R_s(\theta, \epsilon_i)$ and $R_p(\theta, \epsilon_i)$ impart different amplitudes and phases to the TE and TM components that make up a linearly polarized input. In the circular basis $\mathbf{e}_\pm = (\mathbf{e}_{\text{TE}} \pm i\,\mathbf{e}_{\text{TM}})/\sqrt{2}$, this imbalance produces opposite, small centroid shifts for the $\sigma^\pm$ components at nonzero incidence angle $\theta_{\text{inc}}$. The effect is most transparently controlled by the ratio $R_s/R_p$ and by the angular dispersion $\partial_{\theta_{\text{inc}}} \ln R_p$.

For a spatially localized Gaussian probe beam, the electric field amplitudes of the reflected left- and right-handed circularly polarized components are:

$$\mathcal{E}_r^\pm(x_r, y_r, z_r) = \frac{\omega_0}{\omega} \exp\left[-\frac{x_r^2 + y_r^2}{\omega}\right]$$
$$\times \left[R_p - \frac{2ix_r}{k\omega}\frac{\partial R_p}{\partial\theta} \mp \frac{2y_r\cot\theta}{k\omega}(R_s + R_p)\right]. \tag{1}$$

The transverse spatial shift of each spin component due to the PSHE is calculated as:

$$\delta_{p\pm} = \frac{\int y|\mathcal{E}_r^\pm(x_r, y_r, z_r)|^2 \, dx_r dy_r}{\int |\mathcal{E}_r^\pm(x_r, y_r, z_r)|^2 \, dx_r dy_r}. \tag{2}$$

Substituting Eq. (1) into Eq. (2) yields the spin-dependent displacement:

$$\delta_{p\pm} = \mp \frac{k_1\omega_0^2 \operatorname{Re}\left[1 + \frac{R_s}{R_p}\right]\cot\theta_i}{k_1^2\omega_0^2 + \left|\frac{\partial \ln R_p}{\partial\theta_i}\right|^2 + \left|\left(1 + \frac{R_s}{R_p}\right)\cot\theta_i\right|^2}. \tag{3}$$

The shift $\delta_{p+}$ and $\delta_{p-}$ represent the left- and right-circular polarization, respectively, with equal magnitude but opposite direction due to symmetry.

Equation (3) shows explicitly how the PSHE scales with the Fresnel ratio $R_s/R_p$ and the angular derivative $\partial_{\theta_i} \ln R_p$. In our platform, both are tunable because $R_{s,p}$ depend on the intracavity permittivity $\epsilon_2 = 1 + \chi$, which is shaped by the BEC (Sec. III).

### II.B Physical configuration (geometry and conventions)

We next state the optical geometry and modeling conventions used throughout. The probe $(\omega_{\text{P}}, P_{\text{P}})$ impinges on $M_1$ at $\theta_{\text{inc}}$ and is decomposed into TE $(s)$ and TM $(p)$ components relative to the plane of incidence; left/right circular components are defined by $\mathbf{e}_\pm = (\mathbf{e}_{\text{TE}} \pm i\,\mathbf{e}_{\text{TM}})/\sqrt{2}$. The pump $(\omega_{\text{L}}, P_{\text{L}})$ drives the cavity mode coherently. **We employ the standard transfer–matrix formalism to derive the electromagnetic response of the cavity.**

To model this interaction, we use the transfer matrix formalism for the $i^{\text{th}}$ layer of the three-layer system [31,

[32]:

$$\mathcal{M}_i(k_x, \omega_p, d_i) = \begin{pmatrix} \cos(k_z^i) & \frac{i\sin(k_z^i)}{q_{is}} \\ iq_{is}\sin(k_z^i) & \cos(k_z^i) \end{pmatrix}, \qquad (4)$$

where $k_z^i = d_i\sqrt{\epsilon_i k^2 - k^2 \sin^2\theta}$ is the longitudinal wave vector component in the $i^{\text{th}}$ layer of thickness $d_i$, and $q_{is} = \sqrt{\epsilon_i k^2 - k^2 \sin^2\theta}$ characterizes the TE polarization. **The permittivity of the intracavity BEC medium, $\epsilon_2$, is crucial in determining the shift, where $\epsilon_2 = 1 + \chi$ and $\chi$ is the effective susceptibility. To compute $\chi$, we use the quantum Langevin formalism and input-output theory [33], with $\chi$ proportional to the steady-state output field amplitude, i.e., $\chi = E_{\mathbf{T}}$. In the next section, we derive the susceptibility by solving the coupled quantum equations of motion for the intracavity medium system dynamics.**

The complete transfer matrix for the multilayer system is given by:

$$y(k_x, \omega_p) = \mathcal{M}_{1s}(k_x, \omega_p, d_1)\,\mathcal{M}_{2s}(k_x, \omega_p, d_2) \\ \times \mathcal{M}_{3s}(k_x, \omega_p, d_3). \qquad (5)$$

Based on the total transfer matrix formalism given in Eq.(5), the TE-polarized reflection coefficient of the probe field is given by

$$R_s = \frac{q_{1s}(y_{11} + y_{12}q_{3s}) - (q_{3s}y_{22} + y_{21})}{q_{1s}(y_{11} + y_{12}q_{3s}) + (q_{3s}y_{22} + y_{21})}. \qquad (6)$$

Similarly, the TM-polarized reflection coefficient of the probe field is obtained by replacing $q_{is}$ with $p_{im}$, and is given by

$$R_p = \frac{p_{1m}(y_{11} + y_{12}p_{3m}) - (p_{3m}y_{22} + y_{21})}{p_{1m}(y_{11} + y_{12}p_{3m}) + (p_{3m}y_{22} + y_{21})}. \qquad (7)$$

**where $p_{im} = \frac{q_{is}}{\epsilon_i}$ corresponds to the TM counterpart of the transverse wave vector in the $i^{th}$ layer. Note that $y(k_x, \omega_p)$ in Eq. (5) denotes the total transfer matrix and $y_{mn}$ in Eqs. (6)–(7) correspond to its $(m,n)$ element for the TE and TM polarizations, respectively. Once the susceptibility $\chi$ has been evaluated, the reflection coefficients $R_s$ and $R_p$ follow directly from the above equations in combination with the transfer-matrix relations.**

### II.C From susceptibility to Fresnel coefficients and PSHE

The BEC modifies the intracavity response through $\epsilon_2(\omega) = 1 + \chi(\omega)$, which enters the layer matrix $\mathcal{M}_2$ in Eq. (4) and hence the total matrix $y$ in Eq. (5). Consequently, $R_s$ and $R_p$ in Eqs. (6)–(7) become explicit functionals of $\chi(\omega)$ (and of the pump/probe conditions). Because the PSHE shift $\delta_{p\pm}$ [Eq. (3)] depends on the ratio $R_s/R_p$ and on $\partial_{\theta_i}\ln R_p$, tuning the BEC–cavity parameters provides a coherent, *in situ* handle to control the magnitude and sign of the spin-dependent displacement.

TABLE I. Symbols and parameters used. Nominal values match those used in the simulations (see figure captions).

| Symbol | Meaning | Notes/Units |
|---|---|---|
| $\omega_m$ | Bogoliubov mode frequency | $2\pi\times$ 15.2kHz |
| $\kappa$ | Cavity intensity decay rate | $0.05\omega_m$ |
| $\omega_0$ | Probe waist at reflection plane | $50\lambda$ |
| $\lambda$ | wavelength | 780 nm |
| $\chi = E_T$ | Intracavity susceptibility (Sec. III) | complex |
| $\epsilon_2$ | Intracavity permittivity | $1 + \chi$ |
| $R_s, R_p$ | Fresnel reflection (TE/TM) | Eqs. (6)–(7) |
| $q_{is}, p_{im}$ | TE/TM transverse factors | $p_{im} = q_{is}/\epsilon_i$ |
| $y_{mn}$ | Elements of total transfer matrix | Eq. (5) |
| $\delta_{p\pm}$ | PSHE transverse shift | Eq. (3) |

### II.D Perspective and novelty

Conventional PSHE platforms (planar interfaces, metasurfaces, weak-measurement schemes) are typically passive. In contrast, our cavity–BEC architecture offers a *coherently tunable* route to PSHE: the Bogoliubov-mode–mediated susceptibility $\chi$ reshapes $\epsilon_2$, thereby engineering $R_{s,p}$ and the angular dispersion that enter Eq. (3). This coupling enables operating regimes and dynamical control (via pump/probe settings and BEC–cavity parameters) that are not accessible in passive optics, providing a clear perspective on the novelty of our approach.

*Assumptions and validity.* Our analysis uses the paraxial, small-displacement approximation for a Gaussian probe; material dispersion is captured by $\chi(\omega)$ derived in Sec. III. Under the parameters considered, higher-order spatial and nonparaxial corrections are negligible.

### III. System Hamiltonian and Dynamics of a BEC-Cavity Coupled System

We examine a hybrid quantum setup consisting of a BEC coupled to a single–mode optical cavity. The system is governed by the total Hamiltonian:

$$\mathscr{H} = \int dx\,\Psi^\dagger(x)\left[-\frac{\hbar^2}{2m}\frac{d^2}{dx^2} + V_{\text{ext}}(x) + \hbar U_0\cos^2(kx)\,a^\dagger a\right] \\ \Psi(x) + \hbar\omega_a\,a^\dagger a + i\hbar E_L(a^\dagger e^{-i\omega_L t} - a\,e^{i\omega_L t}) \\ + i\hbar E_P(a^\dagger e^{-i\omega_P t} - a\,e^{i\omega_P t}) \qquad (8)$$

Here, $\hat{\Psi}^\dagger(x)$ and $\hat{a}^\dagger$ denote the creation operators for the atomic field and the cavity photons, respectively. The cavity supports a standing–wave mode with spatial profile $\cos(kx)$, where the wave number is $k = 2\pi/\lambda$. The dispersive light shift is quantified by $U_0 = g_0^2/\Delta_a$, with $g_0$ the single–photon atom–cavity coupling and $\Delta_a$ the atom–cavity detuning. Coherent drives at angular frequencies $\omega_L$ and $\omega_P$ enter through the (complex) amplitudes $E_L$ and $E_P$.

Moving into the rotating frame at $\omega_L$ and applying the Bogoliubov approximation for atomic excitations, the

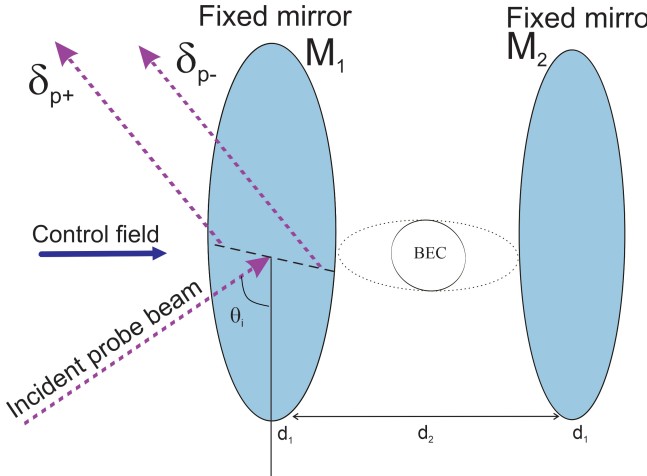

FIG. 1. A schematic of the cavity setup comprises a Bose–Einstein condensate (BEC) of $N$ $^{87}$Rb atoms together with an optical parametric amplifier (OPA). Red arrows indicate the probe field entering and exiting through the left mirror $M_1$ at an incidence angle $\theta_{\text{inc}}$. **The pump field is applied at normal incidence with zero polarization angle (incident angle $= 0°$, polarization $= 0°$).** $\delta_{p+}$ and $\delta_{p-}$ denote the transverse shifts of the reflected probe light corresponding to left- and right-circular polarizations, respectively. Both cavity mirrors, $M_1$ and $M_2$, are kept fixed.

Hamiltonian simplifies to:

$$
\begin{aligned}
\mathcal{H} = &\ \hbar\Delta_a\, a^\dagger a + \hbar\omega_m\, b^\dagger b + \hbar g_{bc}\, a^\dagger a(b + b^\dagger) \\
&+ i\hbar E_L(a^\dagger - a) + i\hbar E_P\left(a^\dagger e^{-i\delta t} - a e^{i\delta t}\right),
\end{aligned}
\tag{9}
$$

where $\Delta_a = \omega_a + \frac{U_0 N}{2} - \omega_L$ includes the mean-field shift from $N$ atoms, $\omega_m = 4\omega_{\text{rec}}$ is the Bogoliubov mode frequency with $\omega_{\text{rec}} = \hbar k^2/2m$, and $g_{bc} = \frac{U_0}{2}\sqrt{N/2}$ represents the effective optomechanical coupling. The final two terms represent the interaction between the cavity and the external driving fields: the pump and probe lasers, characterized by a detuning $\delta = \omega_P - \omega_L$ between the pump and probe frequencies.

Including dissipation and the accompanying quantum fluctuations, the dynamics of the cavity and mechanical modes are governed by the Heisenberg-Langevin equations:

$$
\dot{a} = -(i\Delta_a + \kappa)\, a - i g_{bc}\, a(b + b^\dagger) + E_L + E_P\, e^{-i\delta t}
\tag{10}
$$
$$
+ \sqrt{2\kappa}\, a_{\text{in}},
\tag{11}
$$
$$
\dot{b} = -(i\omega_m + \gamma_m)\, b - i g_{bc}\, a^\dagger a + \sqrt{2\gamma_m}\, \xi,
\tag{12}
$$

where $\kappa$ and $\gamma_m$ denote the damping rates of the cavity and the mechanical resonator, respectively, and $\hat{a}_{\text{in}}$ and $\hat{\xi}$ are the corresponding input–noise operators.

Expanding around steady-state values $a = \alpha + \delta a$, $b = \beta + \delta b$, and retaining only first-order fluctuations yields:

$$
\delta\dot{a} = -(i\Delta + \kappa)\, \delta a - i G_{\text{BC}}\left(\delta b + \delta b^\dagger\right) + E_P\, e^{-i\delta t},
\tag{13}
$$
$$
\delta\dot{b} = -(i\omega_m + \gamma_m)\, \delta b - i G_{\text{BC}}\, \delta a^\dagger - i G_{\text{BC}}^*\, \delta a,
\tag{14}
$$

with $G_{\text{BC}} = g_{bc}|\alpha|$ and $\Delta = \Delta_a - g_{bc}(\beta + \beta^*)$ is the effective detuning.

The steady-state intracavity amplitudes are given by:

$$
\alpha = \frac{E_L(\kappa - i\Delta)}{\kappa^2 + \Delta^2},
\tag{15}
$$

$$
\beta = -\frac{i g_{bc}|\alpha|^2}{i\omega_m + \gamma_m}.
\tag{16}
$$

To study the probe response, we adopt the ansatz:

$$
\delta a(t) = \delta a_- e^{-i\delta t} + \delta a_+ e^{i\delta t},
\tag{17}
$$
$$
\delta b(t) = \delta b_- e^{-i\delta t} + \delta b_+ e^{i\delta t}.
\tag{18}
$$

Substituting into Eqs. (13)–(14) and solving algebraically, the total cavity output field reads:

$$
E_{\text{out}}(t) + E_P e^{-i\delta t} + E_L = \sqrt{2\kappa}\, a(t),
\tag{19}
$$

where the output probe signal is decomposed as:

$$
E_{\text{out}}(t) = E_{\text{out}}^{(0)} + E_{\text{out}}^{(+)} E_P e^{-i\delta t} + E_{\text{out}}^{(-)} E_P e^{i\delta t}.
\tag{20}
$$

The probe transmission is then characterized by:

$$
E_{\text{out}}^{(+)} = \frac{\sqrt{2\kappa}\, a_-}{E_P} - 1,
\tag{21}
$$

$$
E_T = \frac{\sqrt{2\kappa}\, a_-}{E_P},
\tag{22}
$$

where $a_-$ is the intracavity response to the probe field, obtained as:

$$
a_- = \frac{\mathcal{A}}{\mathcal{B}},
\tag{23}
$$

with:

$$
\begin{aligned}
\mathcal{A} = E_P\Big[&2i G_{\text{BC}}^2\, \omega_{\text{m}} + (\gamma_{\text{m}} - i\Delta_p)\,(\gamma_{\text{m}} - i(\Delta_p + 2\omega_{\text{m}})) \\
&(\kappa - i(\Delta_p + 2\omega_{\text{m}}))\Big],
\end{aligned}
\tag{24}
$$
$$
\begin{aligned}
\mathcal{B} = -(\kappa - i\Delta_p)\Big[&-2i G_{\text{BC}}^2\, \omega_{\text{m}} + (\gamma_{\text{m}} - i\Delta_p) \\
&(i\,\gamma_{\text{m}} + \Delta_p + 2\omega_{\text{m}})\,(i\,\kappa + \Delta_p + 2\omega_{\text{m}})\Big] + 2G_{\text{BC}}^2 \\
&(-i\kappa - \Delta_p - 2\omega_{\text{m}})\,\omega_{\text{m}}.
\end{aligned}
\tag{25}
$$

**In the resolved sideband regime, where $\omega_m \gg \kappa$ and the effective detuning is set to $\Delta = \omega_m$, the term $\Delta_p = \delta - \omega_m$ in the previous expression corresponds to the effective probe detuning reduced by the mechanical Bogoliubov mode frequency $\omega_m$,**

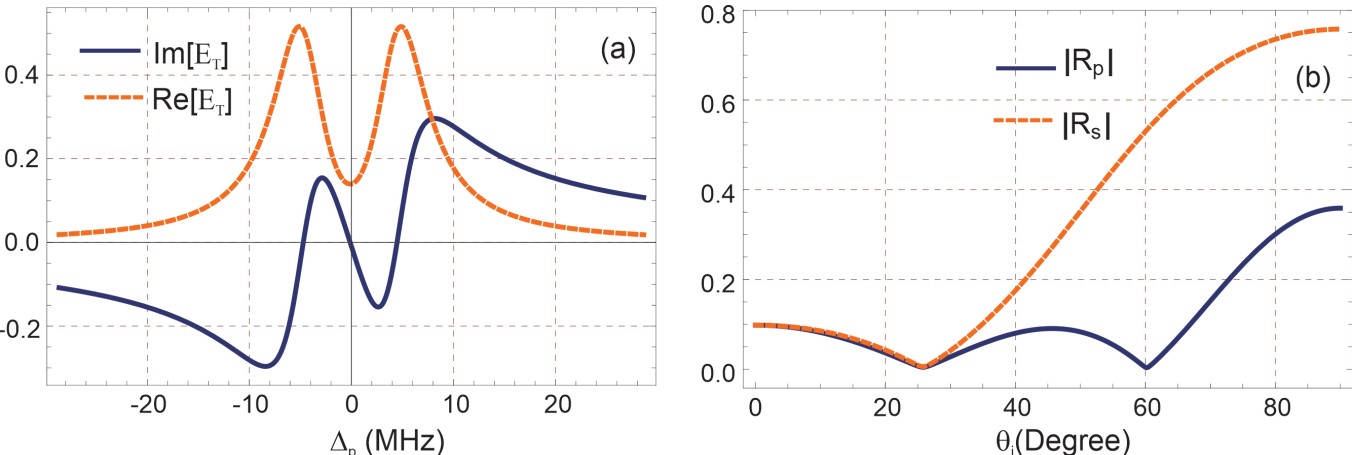

FIG. 2. **(a)** Intracavity BEC response: dispersion (solid) and absorption (dashed) versus probe detuning $\Delta_p$. Parameters: $\omega_m/2\pi = 15.2$ kHz, $\gamma_m/2\pi = 0.21$ kHz, $\omega_{rec}/2\pi = 3.8$ kHz, $\lambda = 780$ nm, $L = 1.25 \times 10^{-4}$ m, $G_{BC} = 0.05\,\omega_m$, and cavity decay $\kappa = 0.05\,\omega_m$.
**(b)** Angular dependence of the Fresnel reflection magnitudes $|R_s|$ and $|R_p|$ as functions of the incidence angle $\theta_i$, evaluated at resonant detuning $\Delta_p = 0$. The remaining parameters are $\epsilon_0 = 1$, $\epsilon_1 = \epsilon_3 = 2.22$, $\epsilon_2 = 1 + E_T$, mirror thicknesses $d_1 = 0.1 \times 10^{-6}$ m and $d_2 = 0.4 \times 10^{-6}$ m.

**i.e., $\Delta_p = \delta - \omega_m$.** The transmission of the probe $E_T$ effectively serves as the susceptibility $\chi$ of the intra-cavity medium, that is, $\chi = \chi_r + i\chi_i \equiv E_T$. Both real and imaginary parts of $\chi$ determine the phase and absorption response of the system, respectively, and can be accessed via homodyne detection techniques. **Spontaneous emission is neglected due to the large atom–cavity detuning, we assume a single linear polarization of the cavity field and consider a scalar (single–component) BEC. The damping rate of the Bogoliubov mechanical mode is introduced phenomenologically to account for residual decay of the collective excitation, and inter-atomic collisions are neglected in the weakly interacting regime.**

## IV. RESULTS AND DISCUSSION

Unless stated otherwise, we adopt experimentally realistic parameters [34, 35]: $\omega_m/2\pi = 15.2$ kHz, $\gamma_m/2\pi = 0.21$ kHz, $\omega_{rec}/2\pi = 3.8$ kHz, $\lambda = 780$ nm, and $L = 1.25 \times 10^{-4}$ m. For the PSHE analysis we further set $\epsilon_0 = 1$, $\epsilon_1 = \epsilon_3 = 2.22$, with mirror-layer thicknesses $d_1 = 0.2\,\mu$m and $d_2 = 5\,\mu$m.

**We operate in the far–off–resonant (dispersive) regime, so that spontaneous emission is negligible and the electronically excited states can be adiabatically eliminated. The analysis is restricted to a single, linearly polarized cavity mode aligned with the $x$–axis. The associated magnetic field is treated implicitly and does not alter the strictly one–dimensional optomechanical motion. Ultracold atom–atom collisions are included at the mean–field level, leading to a renormalization of the effective mechanical frequency; this shift is absorbed into the parameter set used in our sim-**

ulations.

**Only a single internal state of the BEC is considered, since other hyperfine states are far-detuned and remain negligibly populated. The mechanical mode is further characterized by a damping rate $\gamma_m$, which accounts for decoherence arising from atomic collisions as well as residual coupling to the thermal environment.**

Figure 2 summarizes the optical response of the BEC–cavity platform. In panel (a) we display the dispersive part (solid; Re $\chi$) and the absorptive part (dashed; Im $\chi$) as functions of the probe detuning $\Delta_p$. The curves are computed with experimentally accessible parameters: $\omega_m/2\pi = 15.2$ kHz, $\gamma_m/2\pi = 0.21$ kHz, $\omega_{rec}/2\pi = 3.8$ kHz, and a cavity length $L = 1.25 \times 10^{-4}$ m. The light–matter coupling and cavity loss are set to $G_{BC} = 0.05\,\omega_m$ and $\kappa = 0.05\,\omega_m$, respectively.

The dispersion curve exhibits a steep slope in the vicinity of $\Delta_p = 0$, signifying strong group velocity dispersion indicative of slow-light effects. This region corresponds to a rapid variation of the refractive index, a consequence of coherent interference within the hybrid cavity–BEC system. Meanwhile, the absorption profile displays a pronounced transparency window near resonance, suggesting the presence of an electromagnetically induced transparency (EIT)-like phenomenon. This feature stems from destructive interference between competing excitation pathways in the coupled light-matter system, which precisely suppresses absorption at resonance.

In panel (b), we analyze the angular dependence of the Fresnel reflection magnitudes $|R_s|$ and $|R_p|$ at resonant probe detuning $\Delta_p = 0$. The dielectric environment comprises layers with permittivities $\epsilon_0 = 1$, $\epsilon_1 = \epsilon_3 = 2.22$, and $\epsilon_2 = 1 + E_T$, where $E_T$ represents a tunable external control parameter. The multilayer stack includes mirror

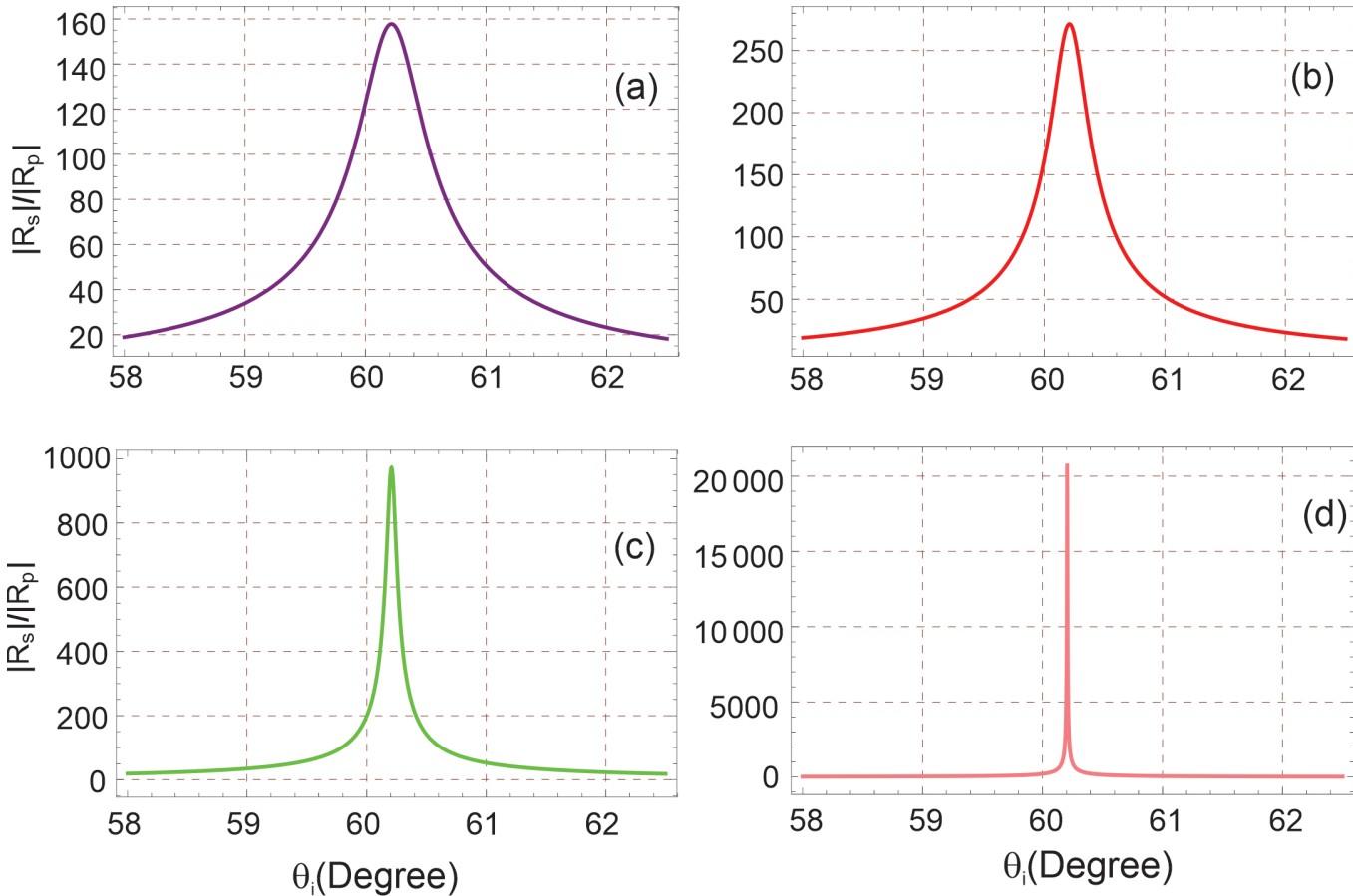

FIG. 3. The amplitude ratio $|R_s|/|R_p|$ is shown as a function of the incidence angle $\theta_i$ for four probe detunings: (a) $\Delta_p = 0$, (b) $\Delta_p = -0.05\,\omega_{\mathrm{m}}$, (c) $\Delta_p = -0.10\,\omega_{\mathrm{m}}$, and (d) $\Delta_p = -0.12\,\omega_{\mathrm{m}}$. Unless specified otherwise, the parameters are $\omega_{\mathrm{m}}/2\pi = 15.2$ kHz, $\gamma_{\mathrm{m}}/2\pi = 0.21$ kHz, $\omega_{\mathrm{rec}}/2\pi = 3.8$ kHz, $\lambda = 780$ nm, $L = 1.25 \times 10^{-4}$ m, $G_{\mathrm{BC}} = 0.05\,\omega_{\mathrm{m}}$, $\kappa = 0.05\,\omega_{\mathrm{m}}$, $\epsilon_0 = 1$, $\epsilon_1 = \epsilon_3 = 2.22$, $\epsilon_2 = 1 + E_T$, with mirror-layer thicknesses $d_1 = 0.1$ $\mu$m and $d_2 = 0.4$ $\mu$m.

thicknesses $d_1 = 0.1 \times 10^{-6}$ m and $d_2 = 0.4 \times 10^{-6}$ m.

The results show that $|R_p|$ displays a dip around the Brewster angle, a classical effect arising from zero reflectivity for $p$-polarized light at a specific angle of incidence. In contrast, $|R_s|$ remains finite and comparatively flat across the angular spectrum, reflecting its insensitivity to such Brewster-angle effects. The sharp angular contrast between the two polarizations establishes a favorable condition for spin-dependent optical phenomena, such as the photonic PSHE, where the angular separation of spin components is maximized.

These findings confirm that the system supports tunable dispersion and polarization-selective reflection, essential for tailoring light propagation, enhancing light–matter interactions, and enabling spin-dependent optical control. The tunability via $\Delta_p$ and $\theta_i$, along with the engineered dielectric environment, underscores the potential of this platform for applications in slow-light devices, precision sensing, and spin-resolved photonic interfaces.

**To quantify the polarization sensitivity, we evaluate the ratio of reflection amplitudes $|R_s|/|R_p|$ for**

**$s-$ and $p-$polarized light as a function of the incidence angle $\theta_i$ for several probe detunings $\Delta_p$ (see Fig. 3). The calculations employ experimentally accessible values: $\omega_{\mathrm{m}}/2\pi = 15.2$ kHz, $\gamma_{\mathrm{m}}/2\pi = 0.21$ kHz, $\omega_{\mathrm{rec}}/2\pi = 3.8$ kHz, and $\lambda = 780$ nm. The optical cavity parameters are $L = 1.25 \times 10^{-4}$ m, $G_{\mathrm{BC}} = 0.05\,\omega_{\mathrm{m}}$, and $\kappa = 0.05\,\omega_{\mathrm{m}}$. The stratified dielectric is specified by $\epsilon_0 = 1$, $\epsilon_1 = \epsilon_3 = 2.22$, and $\epsilon_2 = 1 + E_T$, where $E_T$ represents a tunable contribution set by an external control. The layer thicknesses are fixed at $d_1 = 0.1 \times 10^{-6}$ m and $d_2 = 0.4 \times 10^{-6}$ m.**

**Figure 3(a)–(d) correspond to detuning values $\Delta_p = 0$, $-0.05\omega_{\mathrm{m}}$, $-0.1\omega_{\mathrm{m}}$, and $-0.12\omega_{\mathrm{m}}$, respectively. For $\Delta_p = 0$, the ratio $|R_s|/|R_p|$ exhibits a moderate angular variation around $\theta_i \approx 60.2°$, which serves as the reference case where neither polarization component is preferentially enhanced or suppressed. As the detuning becomes increasingly negative, the angular profile of the ratio becomes noticeably asymmetric. Specifically, for $\Delta_p = -0.05\omega_{\mathrm{m}}$, the ratio increases rapidly**

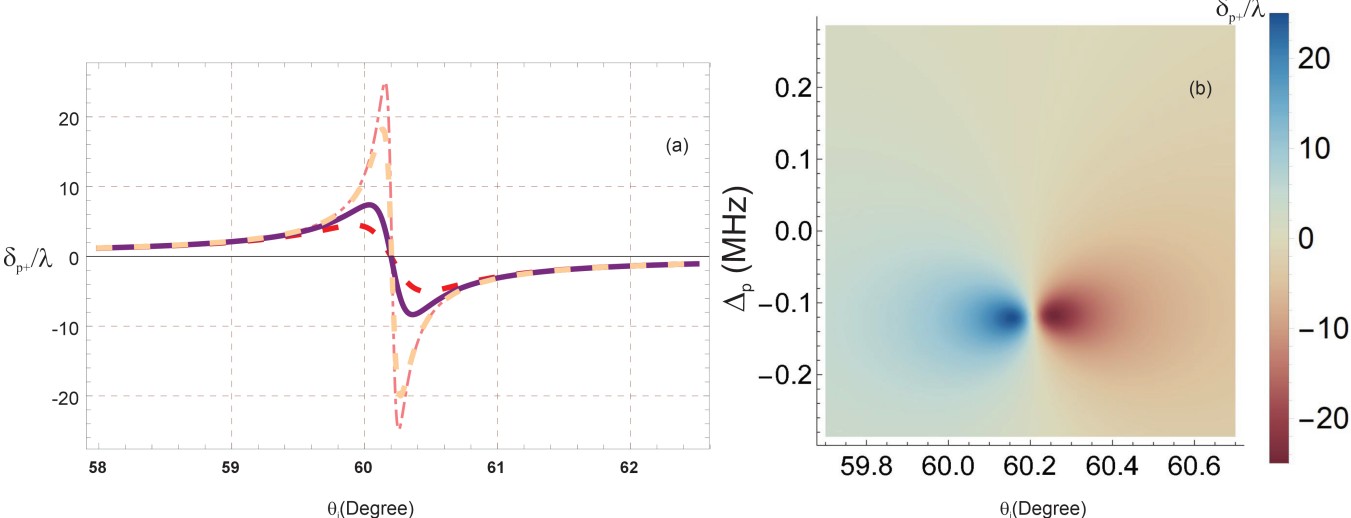

FIG. 4. **(a)** Normalized PSHE displacement $\delta_{p+}/\lambda$ versus incidence angle $\theta_i$ for four probe detunings: $\Delta_p = 0$ (red, dashed), $\Delta_p = -0.05\,\omega_{\mathrm{m}}$ (purple, solid), $\Delta_p = -0.10\,\omega_{\mathrm{m}}$ (light gray, dashed), and $\Delta_p = -0.12\,\omega_{\mathrm{m}}$ (pink, dash–dot). **(b)** Density map of $\delta_{p+}/\lambda$ in the $(\theta_i, \Delta_p)$ plane. Unless stated otherwise, parameters are $\omega_{\mathrm{m}}/2\pi = 15.2$ kHz, $\gamma_{\mathrm{m}}/2\pi = 0.21$ kHz, $\omega_{\mathrm{rec}}/2\pi = 3.8$ kHz, $\lambda = 780$ nm, $L = 1.25 \times 10^{-4}$ m, $G_{\mathrm{BC}} = 0.05\,\omega_{\mathrm{m}}$, $\kappa = 0.05\,\omega_{\mathrm{m}}$, $\epsilon_0 = 1$, $\epsilon_1 = \epsilon_3 = 2.22$, $\epsilon_2 = 1 + E_T$, and mirror–layer thicknesses $d_1 = 0.1\ \mu$m, $d_2 = 0.4\ \mu$m.

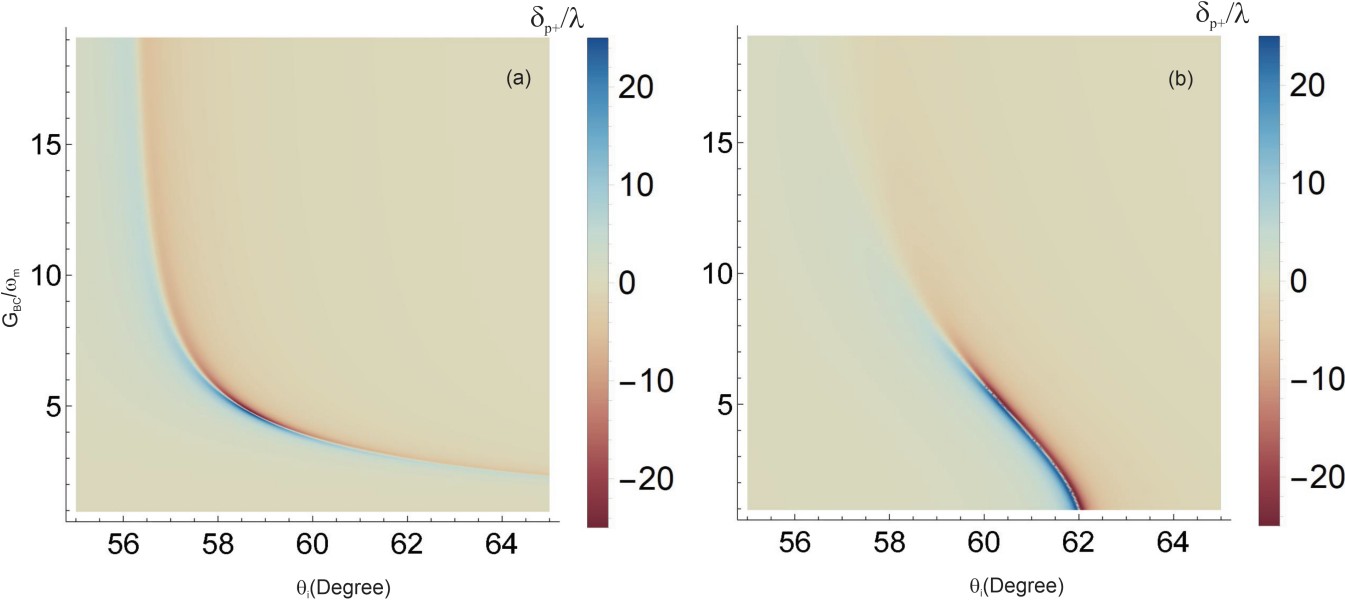

FIG. 5. Density plot of the normalized PSHE shift, $\delta_{p+}/\lambda$, as a function of the incident angle $\theta_i$ and the cavity–BEC coupling strength $G_{\mathrm{BC}}$, for two dissipation regimes: (a) when the cavity dissipation rate $\kappa$ equals the BEC dissipation rate $\gamma_b$, i.e., $\kappa/2\pi = \gamma_m/2\pi = 0.21$ kHz, and (b) when the cavity dissipation rate exceeds that of the BEC, i.e., $\kappa = 0.5\,\omega_m$ and $\gamma_m/2\pi = 0.21$ kHz. The remaining parameters are: probe detuning $\Delta_p = -0.12\,\omega_m$ (corresponding to the maximum PSHE), mechanical frequency $\omega_m/2\pi = 15.2$ kHz, recoil frequency $\omega_{\mathrm{rec}}/2\pi = 3.8$ kHz, wavelength $\lambda = 780$ nm, cavity length $L = 1.25 \times 10^{-4}$ m, dielectric constants $\epsilon_0 = 1$, $\epsilon_1 = \epsilon_3 = 2.22$, $\epsilon_2 = 1 + E_T$, mirror thicknesses $d_1 = 0.1 \times 10^{-6}$ m and $d_2 = 0.4 \times 10^{-6}$ m.

for incidence angles slightly below the reference value and decreases more gradually afterwards.

For larger detuning values, e.g. $\Delta_p = -0.10\omega_{\mathbf{m}}$ and $-0.12\omega_{\mathbf{m}}$, the maximum value of $|R_s|/|R_p|$ increases further, and the angular region where the enhancement occurs becomes narrower around the critical angle. This behavior is associated with the detuning–induced modification of the optical susceptibility, which enhances the coupling of the probe field to spin–polarized modes of the cavity–BEC hybrid system.

These results demonstrate that probe detun-

ing provides a powerful control parameter for tailoring the polarization–dependent reflection. By tuning $\Delta_p$, one can selectively enhance or suppress a given polarization component over a narrow angular range, which may be exploited for polarization–resolved optical filtering and sensing devices.

Figure 4 examines the behavior of the normalized PSHE shift, $\delta_{p+}/\lambda$, as a function of the incident angle $\theta_i$, with particular emphasis on the role of probe detuning $\Delta_p$. Panel (a) shows four representative curves for different detuning values, while panel (b) displays a density map of $\delta_{p+}/\lambda$ in the $(\theta_i, \Delta_p)$ plane.

In Fig. 4(a), the PSHE shift has a pronounced maximum at the critical incident angle $\theta_i \approx 60.2°$. The magnitude and the shape of this maximum are strongly influenced by the probe detuning. For $\Delta_p = 0$, the angular profile is almost symmetric. As the detuning is made negative, the magnitude of the peak increases and the profile becomes asymmetric, with a steeper increase on the low–angle side and a slower decay on the high–angle side. This evolution is a signature of the dispersive response of the optomechanical system, since detuning modifies the effective refractive index and the corresponding spin–orbit coupling strength.

Panel (b) illustrates how $\delta_{p+}/\lambda$ varies across a range of incident angles and detuning values. A distinct sign inversion occurs in the vicinity of the critical angle and zero detuning, highlighting the transition from negative to positive PSHE shift. The blue and red regions in the density plot correspond to opposite transverse shifts of the two circularly–polarized components, and their evolution with detuning reflects the interference between these components. The resulting dip–peak pattern indicates that the spin sensitivity is maximal close to the critical incident angle.

These findings show that the PSHE can be tuned efficiently by adjusting the probe detuning, providing a promising route towards spin–selective photonic devices, such as tunable polarization–dependent beam deflectors and quantum optical sensors.

Figure 5 displays density maps of the normalized PSHE shift $\delta_{p+}/\lambda$ as functions of the incidence angle $\theta_i$ and the BEC–cavity coupling $G_{\rm BC}$, evaluated under two distinct dissipation conditions. These regimes are differentiated by the relative sizes of the cavity decay rate $\kappa$ and the mechanical BEC damping $\gamma_m$. All panels are generated using experimentally realistic parameters specified in the caption.

In panel (a), the condition $\kappa/2\pi = \gamma_m/2\pi = 0.21$ kHz is considered, which signifies balanced dissipation rates for the cavity and the BEC. Under this symmetric dissipation regime, a pronounced peak in the PSHE shift ap-

pears around $\theta_i \approx 59°$, particularly for weak to moderate coupling strengths ($G_{\rm BC}/\omega_m \lesssim 10$). This behavior is attributed to the resonant enhancement of the light–matter interaction, where the PSHE is maximized due to efficient momentum transfer at the interface. The sharp color gradient near this angular region also highlights a strong angular sensitivity of the shift, making the system favorable for precision sensing applications.

In contrast, panel (b) presents the scenario where the cavity dissipation dominates ($\kappa = 0.5\omega_m$, while $\gamma_m/2\pi = 0.21$ kHz remains fixed). In this case, the overall magnitude of the normalized PSHE shift increases compared to panel (a), with broader regions of significant positive values. The peak region shifts slightly to larger incident angles, and the PSHE becomes less sharply localized in both angle and coupling strength. The enhancement in this regime arises due to increased photonic leakage, which facilitates more efficient extraction of spin-dependent components of light, thus amplifying the observable PSHE. However, the trade-off is a reduction in the quality factor of the cavity, which could limit coherence in certain quantum applications.

The comparison between panels (a) and (b) emphasizes the critical role of cavity dissipation in tailoring the PSHE. Specifically, by engineering the relative magnitudes of $\kappa$ and $\gamma_m$, one can modulate the strength and angular profile of the PSHE shift. This tunability is vital for designing reconfigurable photonic devices based on spin–orbit interaction, such as polarization-resolved sensors or optical switches.

Overall, the results confirm that optimal control of dissipation parameters and light–matter coupling enables precise manipulation of spin-dependent photonic transport, with potential applications in chiral quantum optics and topological photonics.

## V. CONCLUSION

We have theoretically examined spin–orbit photonic phenomena in a fixed-mirror optical cavity system coupled to a BEC, where Bogoliubov excitation modes function as effective mechanical resonators. By embedding the BEC within a single-mode cavity, our model uncovers how spin-dependent light–matter interactions manifest as PSHE shifts of a weak probe beam, which are highly sensitive to system parameters.

The analysis reveals strong dispersive and absorptive responses around specific probe detunings, evidencing the role of the BEC–cavity interaction in modifying the system's optical transparency. The angular dependence of the Fresnel coefficients further supports the emergence of spin-selective scattering near the Brewster angle, establishing the angle-dependent origin of the PSHE.

Through systematic exploration of the ratio $|R_s|/|R_p|$ and the corresponding PSHE shift $\delta_{p+}/\lambda$, we demonstrate that the magnitude and sign of the transverse spin-dependent shift are strongly modulated by the probe field detuning. A critical enhancement of the PSHE occurs near resonance, highlighting the cavity-assisted control

over the spin–orbit interaction channel. Additionally, the parameter regimes in which spin-orbit effects are maximized are clearly visualized through two-dimensional contour mappings.

Further insights establish how cavity dissipation governs the SHE response. In particular, under equal dissipation conditions for the cavity and the BEC ($\kappa = \gamma_m$), the shift profile becomes sharper and more localized, while higher cavity losses smooth out the transverse spin shift features. This illustrates the tunability of the spin–orbit effect via dissipative engineering.

Overall, our findings confirm that the interplay between Bogoliubov collective excitations and cavity photon dynamics not only mediates angular momentum exchange but also enables fine control over spin-dependent light steering. The BEC–cavity platform thus serves as a promising medium for developing reconfigurable, miniaturized spin–orbit photonic devices with potential applications in optical sensing, nonreciprocal signal processing, and quantum information routing.

## Acknowledgements

This work was supported by the National Natural Science Foundation of China (Grant No. 12174301), the Natural Science Basic Research Program of Shaanxi (Program No. 2023-JC-JQ-01), and the Fundamental Research Funds for the Central Universities.

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
