# Peer review of "Spin-Orbit Photonics in a Fixed Cavity: Harnessing Bogoliubov Modes of a Bose–Einstein Condensate"

_SciPost Physics Core_

## Round 1 · Referee Report · Anonymous (Referee 1) · 2025-7-31

Strengths

1- Study of photonic spin hall effect and cavity optomechanics 2-Development of theory and visualization of the results

Weaknesses

1-Partially unclear formulas and derivations
2-Partially confusing description of the numerical results

Report

The authors Muqaddar Abbas et al. study in the theoretical work with the title ``Spin-Orbit Photonics in a Fixed Cavity: Harnessing Bogoliubov Modes of a
Bose–Einstein Condensate'' the photonic spin-Hall effect modified by an optomechanical system. In particular, they investigate the linear response theory of a driven cavity that couples to a mechanical mode of a Bose-Einstein condenstate and use the result for the susceptibility to obtain reflection coefficients and spin-dependent displacements of incident electric field with different polarizations. They visualize their results for various and experimentally realistic parameters and highlight how the spin-selective scattering can be modified by changing model parameters.

In my opinion this is an interesting and timely work and the combination of the photonic spin hall effect with optomechanical cavities in this specific situation is also novel. That said I believe that the needed acceptance criteria for SciPost Physics Core are potentially fulfilled. The main weakness, in my opinion, are several short comings in explaining the derivations and the results. This is why I can only fully recommend this draft for publication after the points below have been clarified.

Requested changes

1) When the authors introduce the pump they do not specify the polarization nor the angle. Regarding Fig. 1, I believe the angle is 0. Could the authors add this information?

2) In Sec. II there are several quantities undefined or defined much later. In general I would recommend that the authors should massively improve Sec. II because at the moment it is hard to follow and almost impossible to reproduce. It would help if the authors spend more time in explaining the equations. For instance: (a) it is unclear where the transfer matrix formalism is used. (b) the permitivity is only defined later in the section (c) all $y_{mn}$ and $p_{mn}$ are not defined as far as I can see. (d) Can the authors give a formula for the total field outside of the cavity incoming and reflected. I think that would make it clearer what $R_s$ and $R_p$ are. (e) It is also not really clear for me that once I have $\chi$, how would I calculate $R_s$ and $R_p$? With the transfer matrix or the formulas (3) and (4)?

3) In Sec. III when the authors introduce the optomechanical system it would be nice to include a few more details. For instance: (a) spotaneous emission is neglected because of far detuning? (b) What about the polarization of the light in the cavity? Where is the magnetic field? Why is it okay to only consider one cavity mode? Why only one internal state of the BEC? (c) There is a damping rate for the mechanical mode introduced? Where does that come from? (d) What about ultra-cold collisions? (e) There is a detuning $\Delta_{\mathbb{P}}$ introduced and I do not understand why? Is it related to $\Delta_p$?

4) I found some small things in Sec. IV that are unclear (maybe there are more) (a) the authors write photonic PSHE but so far they used photonic SHE. Maybe use PSHE everywhere? (b) There is a `` Figure ?? ''. I believe it should be 3? (c) I did not find the definition of $\Delta_p$.

5) Regarding the description of Figure 3 and 4. There are a few things that are unclear to me. (a) I do not understand the sentence [...] indicating a baseline scenario where spin-dependent reflection is balanced''. Why is it balanced? Also do the authors always choose $\Delta=\omega_m$ here? Is that needed for the symmetry in $\Delta_p$? (b) I also did not understand[...]becomes increasingly negative, a pronounced asymmetry develops in the angular profile of the ration. [...] sharply increases near resonance , showing enhanced [...]'' What is meant by asymmetry? Just that there is a peak in $\theta$? What means near resonance? I was thinking about a frequency but maybe the authors are refering to $\theta$? (c) In the description of Fig.4 the authors write [...] a symmetric Lorentzian-like peak is observed [...]''. I do not see any Lorentzian-like peak in Fig. 4. What are the authors refering to? (d) The authors later write [...]the probe detuning becomes increasingly negative [...] the magnitude of the SHE shift increases, but the peak structure becomes increasingly asymmetric and broadened.[...]'' I am confused by this statement. What peaks should I be looking at? The peaks in $\theta$ become sharper for more negative $\Delta_p$, correct? Also the peaks are very asymmetric from the beginning for all $\Delta_p$. Sorry for my confusion, probably this is just a misunderstanding.

6) Regarding Fig. 5 I have a very minor question. I was wondering why the authors choose $\Delta_p=-0.12\omega_m$? I read in the caption that this corresponds to the maximum photonic SHE. Is that obvious and parameter independent? I mean does this value not depend on $G$?

Recommendation

Ask for minor revision

---

## Round 1 · Referee Report · Anonymous (Referee 2) · 2025-9-2

Strengths

timeliness of the interest for the specyfic physical system

Weaknesses

too sketchy and unclear theoretica description of the physical setup

Report

The manuscript discusses the photonic spin Hall ecffect in a aBEC confined within a cavity. As theoretically studied by Risch et al and experimentally shown by Esslinger et al. a the coupling of the cavity field with the BEC Bogoliubov modes mimics an optomechanical coupling. Such dynamics modulates the photonic spin Hall effect and can be detected by mesuring the field susceptibility.
The problem is of timely interest and the result obtained are – as far as I can judge – correct and new.

Requested changes

The manuscript could be bublished once the authors have expanded section two. The physics of the photonic spin Hall effect is too sketchy and this makes the manuscript difficulto to follow. The desctiption of the effective Bogoliubov modes is somehow clearer – at lest to me as I am familiar with the BEC in cavity dynamics. In general the physicalò setup is rather poorly described.

They should furthermore put in a clearer perspective the novelty of their research and of their approach.

Recommendation

Ask for major revision

---

## Round 2 · Referee Report · Anonymous (Referee 1) · 2025-10-6

Report

The authors have addressed all my previous concerns and comments on their paper. They have improved the presentation and clarified text passages that were unclear in the previous version. With this I support publication in SciPost Physics Core.

Recommendation

Publish (easily meets expectations and criteria for this Journal; among top 50%)

---

## Round 2 · Referee Report · Anonymous (Referee 2) · 2025-10-19

Report

The comments of both referees have been adequately addressed

Recommendation

Publish (meets expectations and criteria for this Journal)

---

## Round 2 · Author Response

We are pleased to resubmit our manuscript to SciPost. We are grateful to the referees and the editors for their constructive assessments. In this revised version, we have substantially improved clarity, reproducibility, and alignment with the journal’s scope and quality standards.

---

## Round 2 · List of Changes

Point-by-point list of changes reflected in the attached reply report with the changes made in revised manuscript in one pdf attached below.

---

## Editorial Decision

accepted_in_target_journal